# Nonlinear Analysis for a Type-1 Diabetes Model with Focus on T-Cells and Pancreatic *β*-Cells Behavior

**Diana Gamboa** †, **Carlos E. Vázquez** † **and Paul J. Campos** *

Posgrado en Ciencias de la Ingeniería, Tecnológico Nacional de México/I.T. Tijuana, Blvd. Alberto Limón Padilla s/n, Mesa de Otay, 22454 Tijuana, B.C., Mexico; diana.gamboa@tectijuana.edu.mx (D.G.); carlos.vazquez@tectijuana.edu.mx (C.E.V.)
* Correspondence: paul.campos@tectijuana.edu.mx
† These authors contributed equally to this work.

**Abstract:** Type-1 diabetes mellitus (T1DM) is an autoimmune disease that has an impact on mortality due to the destruction of insulin-producing pancreatic *β*-cells in the islets of Langerhans. Over the past few years, the interest in analyzing this type of disease, either in a biological or mathematical sense, has relied on the search for a treatment that guarantees full control of glucose levels. Mathematical models inspired by natural phenomena, are proposed under the prey–predator scheme. T1DM fits in this scheme due to the complicated relationship between pancreatic *β*-cell population growth and leukocyte population growth via the immune response. In this scenario, *β*-cells represent the prey, and leukocytes the predator. This paper studies the global dynamics of T1DM reported by Magombedze et al. in 2010. This model describes the interaction of resting macrophages, activated macrophages, antigen cells, autolytic T-cells, and *β*-cells. Therefore, the localization of compact invariant sets is applied to provide a bounded positive invariant domain in which one can ensure that once the dynamics of the T1DM enter into this domain, they will remain bounded with a maximum and minimum value. Furthermore, we analyzed this model in a closed-loop scenario based on nonlinear control theory, and proposed bases for possible control inputs, complementing the model with them. These entries are based on the existing relationship between cell–cell interaction and the role that they play in the unchaining of a diabetic condition. The closed-loop analysis aims to give a deeper understanding of the impact of autolytic T-cells and the nature of the *β*-cell population interaction with the innate immune system response. This analysis strengthens the proposal, providing a system free of this illness—that is, a condition wherein the pancreatic *β*-cell population holds and there are no antigen cells labeled by the activated macrophages.

**Keywords:** diabetes mellitus; compact invariant sets; nonlinear control

---

## 1. Introduction

Type-1 diabetes mellitus (T1DM) is an autoimmune disease in which, mainly, the production of pancreatic *β*-cells decays at a rate proportional to leukocyte cell growth [1]. This disease is primarily mediated by lymphocyte T-cells, which recognize pancreatic *β*-cells as antigens. The progressive destruction of *β*-cells takes place, leading to a complete loss of insulin production and dysregulation of glucose metabolism. A peak in the global population of individuals under the age of 20 with T1DM or type-2 diabetes is estimated for the year 2045 [2,3]. The development of new treatment options is hampered by our limited understanding of the human pancreas' organogenesis due to the restricted access to primary tissues. However, diabetes is an immunological disease that impacts people worldwide without discrimination of race, sex, nationality, or social status [4]. In recent

decades, substantial increases in diabetes prevalence have demonstrated latency and consistency, with 415 million people worldwide living with diabetes [5].

The International Diabetes Federation estimated that 425 million people in 2017 had T1DM, and as a projection for 2045, that number could reach up to 629 million, which is a global increase of 48% in only 28 years [6]. The average age of prevalence is in the range 20–64 years, which represents 327 million people worldwide; this is a 72% increase over the past 65 years. The global increase from 2017–2045 is overwhelming; a projection of a 35% could increase the cases of T1DM that can escalate to type-2 diabetes in North America and the Caribbean. In the South-Central Americas, there is a projected increase of 62%; meanwhile, European expectations are for a 16% increase in the population with T1DM. The Western Pacific region is the region with the lowest estimated increase; it is projected to be 15%. Southeast Asia, Africa, and the Middle East report higher projections that go from 72% up to 156%, which indicates a state of alarm that needs further research and political strategies. Since the projections are globally overwhelming in terms of how T1DM is increasing over time, robust strategies based on mathematical modeling will improve future health [7]. Models help to explain a system, study the effects of different components, and give predictions about their behavior. Analysis of models via computational and applied mathematical methods is a way to deduce the consequences of the interactions. Moreover, mathematical models allow one to formalize the cause and effect process and relate it to the biological observations. Furthermore, they yield insights into why a system behaves the way it does, thereby providing links between network structure and behavior [8].

At present, there are many studies related to this topic; for instance, mathematical modeling of the glucose–insulin relation was well studied through the years 1989–2012 based on a comparative analysis between the clinical and nonclinical scheme [9], and continues to be studied in a more experimental setting [10]. However, the task of designing therapies [11] that can endorse a reliable path to the minimum suppression set of $\beta$-cells continues to be a primary research issue. Authors in [12] reported a mathematical approach based on the two-day fit modeling of glucose behavior, and they discussed that the parameters played a critical role in allowing the computation of standard tools used as functional insulin therapy, considering the basal rate of insulin and insulin sensitivity factor. Recently, authors in [13] reported the first combined interaction between the variables glucose, insulin, free fatty acids (FFAs), and growth hormone as a set of nonlinear ordinary differential equations.

Nonlinear control theory has proven to be a reliable strategy for presenting results related to a positive bounded invariant domain under the existence of upper bounds, providing a broad understanding with biological implications for a suitable treatment option [14,15]. The study of biological models with nonlinear characteristics is an open invitation for designing a control scheme to establish a statement wherein control inputs contribute to treatment feasibility. Since most of the biological models present prey–predator characteristics, a robust control design enhances the possibility of implementation by applying an embedded system that can consider real-time data [16,17]. The lack of intrinsic robustness in biological systems against various perturbations is a crucial factor that prevents successful stability behavior [18]. The blood glucose control of diabetics can be categorized into open-loop and closed-loop systems, which can feasibly be analyzed by nonlinear theory. Typical diabetic treatment is based on open-loop control in which the patient measures their glucose level during the day and tries to regulate it in a healthy range by injecting an appropriate dose of insulin. This method is not accurate because of the time duration between measurements. Closed-loop control systems present the opposite case because a feedback scheme continually compares the processing data against a reference value, repeating the cycle while searching for the ideal value, as reported in [19–22] and the observer-based nonlinear control design in [23]. Those scientific studies have in common an ordinary differential equation base system that involves the insulin parameter as an input control, where mathematical approaches or computational strategies are applied. However, this type of control analysis is useful when T1DM is part of personal daily care. Nevertheless, transferring the considerations obtained in these mathematical analyses as a possible treatment of diabetes remains a challenge [8].

Recent research suggests a more in-depth development of insulin proliferation due to $\beta$-cells' behavior. In [24], the authors conclude that researchers around the world must continue to monitor trends in type-1 diabetes incidence while working in the areas of prevention, early detection, and improving treatment. Furthermore, in [25], the authors tackled the use of protein biomarkers associated with risk factors in developing cardiovascular diseases when diabetes family antecedents prevail and pass in offspring from the gestational diabetes stage. They conclude that a deeper understanding of the leading causes of diabetes development could improve this topic of research. Therefore, the contribution of this work lies in the mathematical analysis of the complex cellular relation between pancreatic cells when an element of the immune response labels a specific population of cells. Nonlinear theories such as bounded positive invariant domains, localizing compact invariant set, and the Lyapunov method provide a better understanding of how cell–cell relations take place. Thus, we can hopefully define critical parameters that are responsible for the reduction of the pancreatic cell populations, avoiding the high glucose trigger level that may need an insulin control input in the future. The presented results identify some main parameters that could reduce, delay, or avoid levels of non-desired cells, suggesting this as a possible immunotherapy treatment in terms of the variables and parameters of the discussed model.

This paper is organized as follows: The Section 1 is addressed to the literature related to controllers applied in the different models of T1DM, likewise the mathematical theory that is associated with nonlinear models. The Section 2 presents the mathematical model described by a set of fifth-order nonlinear differential equations and the mathematical background necessary to solve the problem of a bounded positive invariant domain existence. Moreover, this section presents the analysis of necessary conditions for stability due to the presence of an invariant plane, and preliminary simulations are shown. The Section 3 presents the nonlinear model with three control inputs, in order to establish asymptotic stability. Finally, Section 4 provides discussion and Section 5 presents conclusions.

## 2. Mathematical Model of T1DM

In 2010, Magombedze et al. proposed a model of the T1DM behavior that consists of five ordinary differential equations. It describes the dynamical correlation between diabetes and immune system response. That is, the interaction of the population sets of resting macrophages ($x_1$), activated macrophages ($x_2$), antigens ($x_3$), autolytic T-cells ($x_4$), and $\beta$-cells ($x_5$), giving the following system [26]:

$$
\begin{aligned}
\dot{x}_1 &= a + (b + k)x_2 - cx_1 - gx_1x_3, \\
\dot{x}_2 &= gx_1x_3 - kx_2, \\
\dot{x}_3 &= lx_2 - mx_3 + qx_4x_5, \\
\dot{x}_4 &= s_T + sx_2x_4 - \mu_Tx_4, \\
\dot{x}_5 &= s_B - qx_4x_5 - \mu_Bx_5.
\end{aligned}
\tag{1}
$$

According to Magombedze et al., the dynamics of each cell population set is as follows: (1) Resting macrophages have a constant supply rate $a$ with a natural death rate $c$ and increase due to the recruitment of activated macrophages with a maximum rate $b + k$, while $g$ is the rate at which resting macrophages become active due to interaction with antigenic cells. (2) Activated macrophages have a supply rate $g$ due to interaction between resting macrophages and antigenic cells, and a natural death rate $k$. (3) Antigen cells increase due to the release of both antigenic peptides by activated macrophages and $\beta$-cell antigenic peptides by dead $\beta$-cells due to the interaction between $\beta$-cells and T-cells with a rate $l$ and $q$, respectively; $m$ is the rate at which antigenic cells are cleared from their population. (4) Autolytic T-cells have a constant supply rate $s_T$ and a natural death rate $\mu_T$ with proliferation rate $s$ due to a profile of cytokines and chemokines, induced by activated macrophages. (5) $\beta$-Cells have a constant supply rate $s_B$ and a natural death rate $\mu_B$, while $q$ is the depletion of the $\beta$-cell population due to interactions between $\beta$-cells and T-cells. The description of each parameter and its estimation

are taken from [26] and summarized in Table 1. Parameters marked with $(*)$ are critical values in the islet regeneration region in the pancreas [27].

**Table 1.** Parameter descriptions, values, and units.

| Parameter | Definition | Value | Units |
|---|---|---|---|
| $a$ | Macrophage supply | 50 | $\text{mm}^{-3}\,\text{day}^{-1}$ |
| $b$ | Macrophage induced supply | 0.3 | $\text{day}^{-1}$ |
| $c$ | Macrophage death rate | 0.1 | $\text{mm}^{-3}\,\text{day}^{-1}$ |
| $g$ | Rate of antigen uptake | $65 \times 10^{-6}$ | $\text{day}^{-1}$ |
| $k$ | Macrophage deactivation | 0.2 | $\text{day}^{-1}$ |
| $l*$ | Induced $\beta$-cell damage | $250 \times 10^{-6}$ | $\text{day}^{-1}$ |
| $m$ | Decay rate of $\beta$-cell proteins | 0.025 | $\text{day}^{-1}$ |
| $q$ | Damage of autolytic cells on $\beta$-cells | $2 \times 10^{-6}$ | $\text{mm}^{-3}\,\text{day}^{-1}$ |
| $s_T*$ | Supply of autolytic cells | 20 | $\text{mm}^{-3}\,\text{day}^{-1}$ |
| $s$ | Proliferation of autolytic T-cells | $2 \times 10^{-5}$ | $\text{day}^{-1}$ |
| $\mu_T$ | Death rate of T-cells | 0.02 | $\text{day}^{-1}$ |
| $s_B*$ | Supply of $\beta$-cells | 20 | $\text{mm}^{-3}\,\text{day}^{-1}$ |
| $\mu_B$ | Death rate of $\beta$-cells | 0.02 | $\text{mm}^{-3}\,\text{day}^{-1}$ |

## 2.1. Localization of Compact Invariant Sets (LCIS)

### 2.1.1. Mathematical Preliminaries

The general method of LCIS is applied to determine the location of a domain including all compact invariant sets of differential equation systems. This method is useful in cases which are necessary to understand the long-term behavior of a dynamical system. Now, consider a nonlinear system represented as follows:

$$\dot{x} = f(x), \tag{2}$$

where $x \in \mathbb{R}^n$, $f(x) = (f_1(x), \ldots, f_n(x))^T$ is a differentiable vector field. Let $h(x) \in C^\infty(\mathbb{R}^n)$ be a function such that $h$ is not the first integral of the system (2). The function $h$ is exploited in the solution of the localization problem of compact invariant sets and it is called a localizing function. $h|_U$ denotes the restriction of $h$ on a set $U \subset \mathbb{R}^n$. $S(h)$ denotes the set $\{x \in \mathbb{R}^n \mid L_f h(x) = 0\}$, whereas $L_f h(x)$ is the Lie derivative in the vector field of $f(x)$. In order to determine the localizing set, it is necessary to define $h_{\inf}(U) := \inf\{h(x) \mid x \in U \cap S(h)\}$ and $h_{\sup}(U) := \sup\{h(x) \mid x \in U \cap S(h)\}$. Therefore, for any $h(x) \in C^\infty(\mathbb{R}^n)$ all compact invariant sets of the system (2) located in $U$ are contained in the set $K(U; h)$, defined as $\{x \in U \mid h_{\inf}(U) \le h(x) \le h_{\sup}(U)\}$, and if $U \cap S(h) = \varnothing$, there are no compact invariant sets located in $U$ [28,29].

### 2.1.2. Mathematical Development

In this section we compute the domain of attraction containing all compact invariant sets of the system (1), making it possible to determine lower and upper bounds. Bounds are defined by inequalities depending on the system's parameters, giving a global insight about the ultimate densities of each cell population in long time intervals; thus, in the biological sense, these mathematical assumptions define the minimum and maximum carrying capacity of the cell population. The achievement of the bounded positive invariant domain (BPID) is possible when all upper bounds of (1) cross each other as a result of applying the LCIS method.

The BPID establishes that if all trajectories of a system enter the positive invariant domain, they remain within it all the time. Notice that obtaining BPID for the system (1) defined in the $\mathbb{R}^5_{0,+}$ orthant, which contains all state variables of the system under analysis, cannot be achieved due to the complexity of the system even when the system satisfies positiveness—that is, even if all the state variables are considered positive $(x_n > 0)$. In this particular case, BPID is possible if the planes $x_1$ and

$x_4$ are equal to zero. Nevertheless, this consideration is biologically meaningless since $x_1$ represents resting macrophages, an essential population set in the immunological response against any foreign or internal antigen. On the other hand, an analysis considering the invariant plane $x_4 = 0$ implies a low response of the autolytic T-cells; therefore, the activation of lymphocyte cytotoxic T-cells is null. Autolytic T-cells represent a subset of helper T-cells, and in the scheme of proposing a suppression of the immunological response, it implies the possibility of a feasible treatment. Thereby, the mathematical implications lead to the idea that control input is necessary.

## 2.2. LCIS for the Invariant Plane $\mathbb{R}^5_+ \cap \{x_4 = 0\}$

It is essential to mention that $x_4$ represents a critical variable that leads to the activation of lymphocyte cytotoxic T-cells with the help of autolytic T-cells, which pursue the elimination of an antigen. In this scenario, the antigen represents $\beta$-cells; once activated, macrophages label them. Hence, in this work we assumed that the invariant plane can hypothetically be considered as a control input representing a feasible scheme to pose a treatment parameter that leads to the non-activation of this cell population. Under this consideration, the system represented by (1) becomes:

$$
\begin{aligned}
\dot{x}_1 &= a + (b + k)x_2 - cx_1 - gx_1x_3, \\
\dot{x}_2 &= gx_1x_3 - kx_2, \\
\dot{x}_3 &= lx_2 - mx_3, \\
\dot{x}_5 &= s_B - \mu_B x_5.
\end{aligned}
\tag{3}
$$

**Theorem 1.** *The localization of all compact invariant sets is achieved under the restriction of the invariant plane $x_4$ by applying localizing functions defined by the variables of the system (3), in order to close a domain bounded with upper bounds. The domain of interest is obtained as the set $K = x_{1\,\max} \cap x_{2\,\max} \cap x_{5\,\max} \cap x_{3\,\max} \cap x_4 = 0$, if and only if the conditions (4)–(6) are satisfied under the restriction of $\mathbb{R}^5_{0,+}$ due to biological implications.*

**Proof.** The following localizing function $h_1 = x_1 + x_2 - \beta_1 x_3 + x_5$ is proposed, and after substituting the model (3) equation, where $\beta_1$ is a free positive parameter, the set $h_1 \mid_{S(h_1)}$ is held under the following conditions:

$$
\begin{aligned}
c &\geq m, & (4) \\
\beta_1 &\geq \frac{b + m}{l}, & (5) \\
\mu_B &\geq m, & (6)
\end{aligned}
$$

then, the set $K(h_1)$ exists in the positive orthant that contains the upper bounds for the variables $x_1$, $x_2(t)$, $x_5(t)$. In order to establish an upper bound for $x_3$, we propose the localizing function $h_2 = x_3$; therefore, the set $K(h_2)$ is obtained by applying the iterative theorem [29]. Hence, the set $K_1 = K(h_1) \cap K(h_2) \mid_{K(h_1)}$ that contains all compact invariant sets of model (3) is represented as:

$$
K_1 = \left\{ x_{1\,\max} + x_{2\,\max} - \beta_1 x_3 + x_{5\,\max} \leq \frac{a + s_B}{m} \right\} \cap \left\{ x_{3\,\max} \leq \frac{lx_{2\,\max}}{m} := \frac{l(a + s_B)}{m^2} \right\}.
\tag{7}
$$

□

The BPID defined by the invariant plane $x_4 = 0$ denoted by (7) establishes a leading path to the hypothesis in which a control input may guarantee the non-activation of the autolytic T-cells despite a population set of activated macrophages that are demanding a high response from autolytic T-cells to activate lymphocyte T-cytotoxic cells as an innate response. As a consequence, the $\beta$-cell population that is labeled as an antigen by activated macrophages will have less volume compared with the

population set of $\beta$-cells that continue the function of producing insulin. Therefore, the mathematical analysis of the invariant plane exhibits the potential existence of immunotherapy that can counteract the growth rate in which $\beta$-cells come labeled as antigens.

*2.3. LCIS for $\mathbb{R}_+^5 \cap \{x_4 > 0\}$*

In this subsection, the mathematical analysis tackles the scheme when autolytic T-cells are activated and cannot deactivate their primary function, the activation of the lymphocyte cytotoxic T-cells. In [26], the authors reported a proliferation of T-cells due to the profile of cytokines and chemokines induced by activated macrophages ($sx_2x_4$), and also estimated the population for each set of cells. Therefore, the method of LCIS contributes to obtain the maximum carrying capacity that all cell sets will have in a long time period, independently of how they behave. Hence, let us propose some localizing functions in a way that can mathematically express the maximum and lower carrying capacity.

In order to define the maximum or minimum density of resting macrophages, activated macrophages, and antigen cells, consider the localizing function $h_3 = x_1 + x_2 - \beta x_3$, wherein $\beta$ is a free parameter, while its Lie derivative is contained in the set $S(h_3)$ expressed in the form $S(h_3) = \{(\beta l - b)x_2 = a - c_1 x_1 - \beta q x_5 x_4 + \beta m x_3\}$. Then, the set $h\mid_{S(h_3)}$ is defined as:

$$h\mid_{S_{(3)}} = \left(1 - \frac{c}{\beta l - b}\right)x_1 + \frac{a}{\beta l - b} - \frac{\beta q}{\beta l - b}x_5 x_4 + \beta\left(\frac{m}{\beta l - b} - 1\right)x_3, \tag{8}$$

where the $\beta$ domain is

$$\frac{(m+b)}{l} \leq \beta \leq \frac{(c+b)}{l}, \tag{9}$$

while the upper bound for the set $K_3(h_3)$ is as follows:

$$K_3(h_3) = \left\{h_1 \leq h_1\mid_{S(h_1)} := \frac{a}{\beta l - b}\right\}. \tag{10}$$

Hence, the upper bounds for resting and activated macrophages are given by

$$K_3(h_3) = \left\{x_1 \leq x_{1\max} := \frac{a}{\beta l - b}; x_2 \leq x_{2\max} := \frac{a}{\beta l - b}\right\}. \tag{11}$$

The ultimate density of the $\beta$-cell population is determined by a second localizing function $h_4 = x_5$, where the set $S(h_4)$ is defined as $S(h_4) = \{\mu_B x_5 = s_B - q x_5 x_4\}$, leading to a set $K(h_4)$ which defines the maximum cell population by

$$K(h_4) = \left\{x_5 \leq x_{5\max} := \frac{s_B}{\mu_B}\right\}. \tag{12}$$

A third localizing function $h_5 = x_3 + x_5$ is used to determine the maximum population set of antigen cells, leading to the set $S(h_5)$ defined as $S(h_5) = \left\{x_5 = \frac{l}{\mu_B}x_2 - \frac{m}{\mu_B}x_3 + \frac{s_B}{\mu_B}\right\}$, thus $h_5 \leq h_5\mid_{S(h_5)}$ under the condition $m \geq \mu_B$, implying that there exists an upper bound for the population of antigen cells. The set $K(h_5)$ is achieved under the restriction of the set $K(h_3)$ as

$$K(h_5) \cap K(h_3) = \left\{h_3 \leq h_3\mid_{S(h_3)} := \frac{s_B}{\mu_B} + \frac{1}{\mu_B}x_{2\max}\right\}. \tag{13}$$

Furthermore, a localizing function $h_6 = x_4$ is defined to obtain the carrying capacity of T-cells, given a set $S(h_6)$ defined by

$$S(h_6) = \left\{x_4 = \frac{s_T}{\mu_T - sx_2}\right\}, \tag{14}$$

wherein an upper bound of T-cell exists if $\beta$ fulfils the condition (9), (11), and $x_2 < \frac{\mu_T}{s}$ is satisfied, as long as

$$0 < \frac{a}{\beta l - b} < \frac{\mu_T}{s}, \tag{15}$$

so that the free parameter $\beta$ is held under the intersection

$$\beta := \left\{ \left\{ 0 < \frac{a}{\beta l - b} \right\} \cap \left\{ \frac{a}{\beta l - b} < \frac{\mu_T}{s} \right\} \right\}. \tag{16}$$

Hence, the set $K(h_6)$ is defined as follows:

$$K(h_6) = \left\{ x_4 \leq x_{4\,\mathrm{max}} := \frac{s_T}{\mu_T - s x_{2\,\mathrm{max}}} \right\}. \tag{17}$$

Additionally, throughout the localizing function $h_6$ the lower bound of these cells can be obtained. In this case, the function $S(h_6)$ may also be presented as

$$S(h_6) = \{ s_T = \mu_T x_4 - s x_2 x_4 \}, \tag{18}$$

thus, the set $K(h_6)$ is defined as

$$K_6(h_6) = \left\{ \frac{s_T}{\mu_T} := x_{4\,\mathrm{min}} \leq h_6 \leq x_{4\,\mathrm{max}} := \frac{s_T}{\mu_T - s x_{2\,\mathrm{max}}} \right\}. \tag{19}$$

Therefore, if conditions (9), (15), and (16) are satisfied, then the BPID existence for system (1) is achieved and contained inside of the following domain:

$$K_2 = \{ x_{1\,\mathrm{max}} \cap x_{2\,\mathrm{max}} \cap x_{3\,\mathrm{max}} \cap x_{5\,\mathrm{max}} \} \subset \{ x_{4\,\mathrm{min}} \leq x_4 \leq x_{4\,\mathrm{max}} \}. \tag{20}$$

A summary of lower and upper bounds for each cell population can seen in Table 2, complementing by quantitative cell sets those qualitative results presented in [26].

**Table 2.** Upper and lower bounds.

| Localizing Functions | Conditions | Localizing Set |
|---|---|---|
| $h_3 = x_1 + x_2 - \beta x_3$ | $\frac{(m+b)}{l} \leq \beta \leq \frac{(c+b)}{l}$ | $K(h_3) = \left\{ h_3 \leq h_3 \mid_{S(h_3)} := \frac{a}{\beta l - b} \right\}$ |
| $h_4 = x_5$ | | $K(h_4) = \left\{ x_5 \leq x_{5\,\mathrm{max}} := \frac{s_B}{\mu_B} \right\}$ |
| $h_5 = x_3 + x_5$ | $m \geq \mu_B$ | $K(h_5) \cap K(h_3) = \left\{ h_5 \leq h_5 \mid_{S(h_5)} := \frac{s_B}{\mu_B} + \frac{1}{\mu_B} x_{2\,\mathrm{max}} \right\}$ |
| $h_6 = x_4$ | | $K(h_6) = \{ x_{4\,\mathrm{min}} \leq h_6 \leq x_{4\,\mathrm{max}} \}$ |

The case of a BPID with $x_4 > 0$ includes all trajectories of the system and tends to the only non-negative and non-zero equilibrium point, satisfying the biological sense. That is, the cell populations of resting macrophages and active macrophages depend on the decay rate of $\beta$-cell protein and the macrophages' death rate values, respectively. However, the decay rate of $\beta$-cell protein must be greater than or equal to the natural death rate of $\beta$-cell. Therefore, these conditions exploit a closed-loop strategy to prove that the hypothesis is valid based on nonlinear control. This condition reinforces the hypothesis made when invariant plane $x_4$ holds. Conditions (16) and $m \geq \mu_B$ establish the potential feasibility of suppressing at least two main variables of the model, as part of the immunological response, by control inputs. Nevertheless, these inputs in a closed-loop design aim at suppressing the innate response by a combined population of both activated macrophages and autolytic T-cells, avoiding the creation of antigen cells by $\beta$-cells.

## 3. Nonlinear Controller Design

The $\beta$-cell population triggers insulin markers to maintain normal blood glucose levels by regulating carbohydrate lipid and protein metabolism through its mitogenic effects via blood vessels. The only test that can provide a direct measurement of the $\beta$-cell population set is the C-peptide test [30], which has proven to be a reliable clinical test, despite measurement risk factors that depend on the level of expertise of the clinician. This type of analysis depends on human estimation and numerical assumptions [31]. The C-peptide test is a useful indicator of $\beta$-cell function measurement that allows direct discrimination between insulin sufficiency and insulin deficiency in individuals with T1DM. In this case, we propose three control inputs for the variables related to activated macrophages, $\beta$-cells presented as antigens and autolytic T-cells. As previously discussed, this is derived as the biological implication due to a population set of resting macrophages ($x_1$) that label a population set of $\beta$-cells ($x_5$) as antigens ($x_3$), thereby $x_1$ becomes activated macrophages ($x_2$) and directly responsible for autolytic T-cell ($x_4$) stimulation, wherein their primary function is to trigger the lymphocytes' cytotoxic T-cells response. The following control hypothesis aims to prevent diabetes by reinforcing the immunological response; hence, inhibiting the $x_2$ responses would contribute to a decreasing $x_5$ labeled as $x_3$, and as a consequence, it will not be eliminated. Therefore, it is necessary to control the populations of these undesired cells to avoid both the destruction of $x_5$ once the antigen tag is made and an activation of lymphocyte cytotoxic T-cells is called for by the immune response.

In accordance with the aforementioned, system (1) can be expressed in closed-loop control form as follows:

$$
\begin{aligned}
\dot{x}_1 &= a + (b+k)x_2 - cx_1 - gx_1x_3, \\
\dot{x}_2 &= gx_1x_3 - kx_2 + u_1, \\
\dot{x}_3 &= lx_2 + qx_5x_4 - mx_3 + u_2, \\
\dot{x}_4 &= s_T + sx_2x_4 - \mu_Tx_4 + u_3, \\
\dot{x}_5 &= s_B - qx_5x_4 - \mu_Bx_5,
\end{aligned}
\tag{21}
$$

where $u_1$, $u_2$, and $u_3$ are control inputs that in a biological sense have the objective of preserving $\beta$-cell population in pancreas islets. Then, to determine the conditions for each control input, we consider the following Lyapunov candidate function:

$$
V = \frac{1}{2}\sum_{i=1}^{5}\beta_i x_i^2,
\tag{22}
$$

where its derivative is $\dot{V} = \sum_{i=1}^{5}\beta_i x_i \dot{x}_i$, $\beta_i \in \mathbb{R}_+^5$ with $i \in \{\mathbb{Z}^+ \le 5\}$ are free positive parameters, and after substituting each $\dot{x}_i \in \mathbb{R}_+^5$ of system (21) into the derivative of (22), $\dot{V}$ is defined by

$$
\begin{aligned}
\dot{V} = {} & \beta_1\left(ax_1 + (b+k)x_1x_2 - cx_1^2 - gx_1^2x_3\right) + \beta_2\left(gx_1x_2x_3 - kx_2^2 + u_1x_2\right), \\
& + \beta_3\left(lx_2x_3 + qx_3x_4x_5 - mx_3^2 + u_2x_3\right) + \beta_4\left(s_Tx_4 + sx_2x_4^2 - \mu_Tx_4^2 + u_3x_4\right), \\
& + \beta_5\left(s_Bx_5 - x_5^2x_4 - \mu_Bx_5^2\right).
\end{aligned}
\tag{23}
$$

By inspection of Equation (23), it can be seen that three control inputs may not be necessary and stability conditions could be satisfied by less than three of them. Nevertheless, in [26] it is established that if the magnitudes of parameters $\{a, g, q, s, s_T, s_B\}$ decrease, it could be a positive result towards the control of diabetes in early diagnostics. On the other hand, if the control inputs parametrization contains some of the parameters defined by $\{c, k, m, \mu_B, \mu_T\}$, it means that a diabetic condition is evident and the disease needs to be treated. Therefore, less than three control inputs implies the

necessity of a parameter combination of both sets, meaning that a patient could be under a diabetes condition. Thereby, the following proposition aims to define the entries based on the parameters that have a direct impact on preventing diabetes. Hence, the control inputs are as follows:

$$u_1 = -gx_1x_3, \tag{24}$$

$$u_2 = -lx_2 - qx_5x_4, \tag{25}$$

$$u_3 = -sx_2x_4. \tag{26}$$

Substituting the control inputs (24)–(26) into (23) and completing the quadratic form for variables $x_2$, $x_4$, and $x_5$ gives that $\dot{V}$ is as follows:

$$
\begin{aligned}
\dot{V} = {} & -\beta_2 k \left( x_2 - \frac{\beta_1(b+k)x_1}{2\beta_2 k} \right)^2 + \frac{\beta_1^2(b+k)^2 x_1^2}{4\beta_2 k} - \beta_4 \mu_T \left( x_4 - \frac{s_T}{2\mu_T} \right)^2 + \frac{\beta_4 s_T^2}{4\mu_T}, \\
& -\beta_5 \mu_B \left( x_5 - \frac{s_B}{2\mu_B} \right)^2 + \frac{\beta_5 s_B^2}{4\mu_B} - \beta_1 c x_1^2 + \beta_1 a x_1 - \beta_3 m x_3^2 - \left( \beta_1 g x_1^2 x_3 + \beta_5 x_5^2 x_4 \right).
\end{aligned} \tag{27}
$$

Now, completing the quadratic form for the positive terms that contain $x_1$ and factorizing the common term $-\beta_1$, the equation (27) is defined as

$$
\begin{aligned}
\dot{V} = {} & -\beta_2 k \left( x_2 - \frac{\beta_1(b+k)x_1}{2\beta_2 k} \right)^2 - \beta_4 \mu_T \left( x_4 - \frac{s_T}{2\mu_T} \right)^2 - \beta_5 \mu_B \left( x_5 - \frac{s_B}{2\mu_B} \right)^2 \\
& -\beta_1 A \left( x_1 - \frac{a}{2A} \right)^2 - \beta_3 m x_3^2 - \left( \beta_1 g x_1^2 x_3 + \beta_5 x_5^2 x_4 \right) + \frac{\beta_1 a^2}{4A} + \frac{\beta_4 s_T^2}{4\mu_T} + \frac{\beta_5 s_B^2}{4\mu_B},
\end{aligned} \tag{28}
$$

with

$$A = c - \frac{\beta_1(b+k)^2}{4\beta_2 k}, \tag{29}$$

where the following condition for $\beta_2$ must be satisfied to guarantee the positiveness of $A$:

$$\beta_2 > \frac{\beta_1(b+k)^2}{4ck}. \tag{30}$$

Hence, since all variables present nonlinear dynamics in the positive orthant due to their biological implications, and as the analysis demonstrates in the previous section, Equation (22) satisfies Lyapunov asymptotic stability if the following inequality is also satisfied:

$$
\begin{aligned}
& \beta_1 A \left( x_1 - \frac{a}{2A} \right)^2 + \beta_2 k \left( x_2 - \frac{\beta_1(b+k)x_1}{2\beta_2 k} \right)^2 + \beta_3 m x_3^2 \\
& + \beta_4 \mu_T \left( x_4 - \frac{s_T}{2\mu_T} \right)^2 + \beta_5 \mu_B \left( x_5 - \frac{s_B}{2\mu_B} \right)^2 > B := \frac{\beta_1 a^2}{4A} + \frac{\beta_4 s_T^2}{4\mu_T} + \frac{\beta_5 s_B^2}{4\mu_B}.
\end{aligned} \tag{31}
$$

The proposed control inputs are biologically sound. That is, these involve the parameters that trigger the progression of diabetes and those cell populations that influence the diagnosis of this disease, also supported by the stability analysis. However, a physical implementation is still a challenge because it is not possible to modify cell populations such as resting macrophages, since they are produced naturally by the immune system. Hence, if a treatment or a form of intervention can reduce such parameters, it could give positive results in diabetes prevention.

*Numerical Simulations*

This section presents numerical simulations. The simulation setting is according to the parameter values given in Table 1. Figure 1 shows the BPID construct with the invariant plane $K_1$. As can be seen, when there is no set of T-cells concentration into the system, regardless of the initial populations of the cells that remain inside it, they will tend to their equilibrium level, which means a stable condition of the disease where there is no progression. Therefore, it can be concluded that the population of $\beta$-cells in the pancreas is optimal. In this case, this simulation is considered valuable due to the feasibility of proposing at least one control input for the system (1) that could help to ensure a stable population of T-cells, or at least prevent their indiscriminate spread. Moreover, the invariant plane of $x_4 = 0$ implies that there is no immunological response. Therefore, all cell populations tend to their optimal concentrations. Figure 2a,b presents the upper bound domain of the localizing set $K(h_3)$ for both resting and activated macrophages; Figure 2c presents the upper bound domain of the localizing set $K(h_5)$ for antigen cells concentration, demonstrating that the variables associated with activated macrophages have immediate response once the $\beta$-cells are presented as antigen. It is a natural response, since they influence the lymphocyte cells by activating the autolytic T-cells. Figure 2d presents the upper and lower bounds for the T-cell population as a result of the localizing set $K(h_6)$. As can be seen from the localizing set, "it is important to highlight that exists a minimum level of a cell population that has a direct impact on triggering cytotoxic T-cells," Figure 2e presents the upper bound for the $\beta$-cell population resulting from the localizing set $K(h_4)$. Therefore, this analysis exposes the complex interaction between these cell populations, demonstrating that when there are no activated macrophages that label a population set of $\beta$-cells as antigens, there is no autolytic response required. This leads to the development of a mathematical proposal supported by nonlinear control theory to treat this disease.

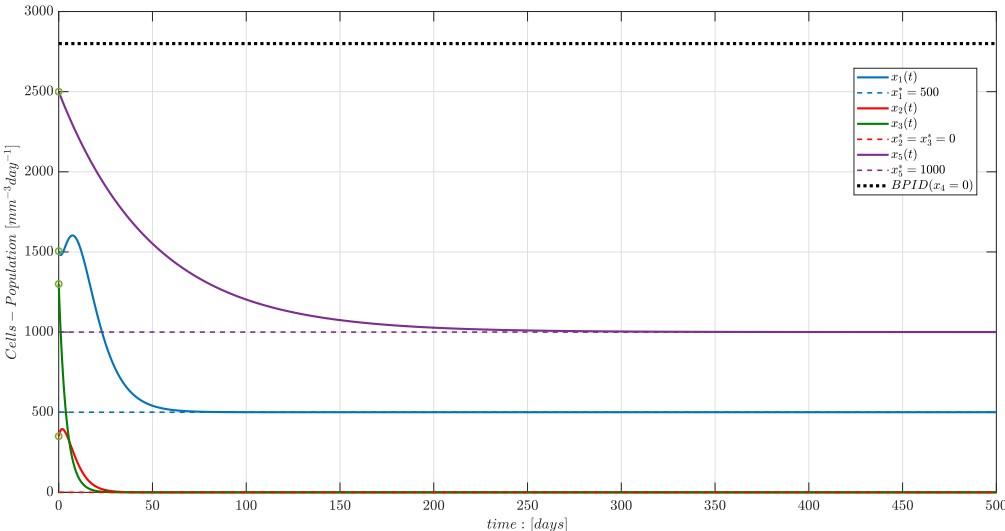

**Figure 1.** Upper bound for all cell population sets by the bounded positive invariant domain (BPID) set $K_1$, considering the invariant plane $x_4(t) = 0$.

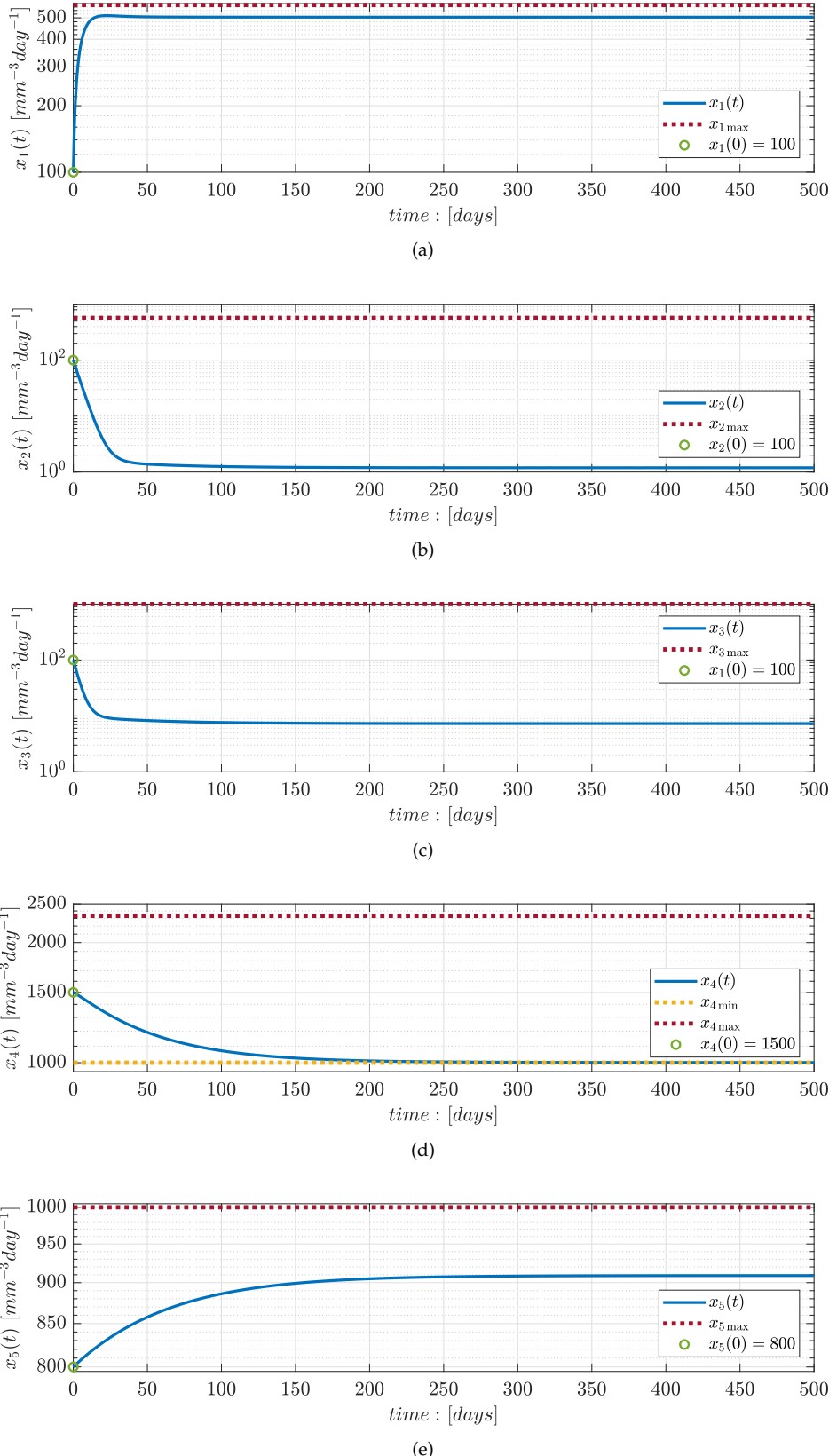

**Figure 2.** Upper and lower bounds for each cell populations. (**a**) Maximum concentration of resting macrophage cells. (**b**) Maximum concentration of activated macrophage cells. (**c**) Maximum concentration of antigen cells. (**d**) Minimum and maximum concentrations of T-cells. (**e**) Maximum concentration of $\beta$-cells.

Figure [3] presents the convergence of the cell populations to their desired cell concentrations for resting macrophages, activated macrophages, and antigens. Figure [4] presents the convergence of the cell populations to the desired cell concentrations for T-cells and $\beta$-cells due to the control actions (24), (25), and (26). In both figures, the solid line, dotted line, and dashed line represent the open-loop natural response, the closed-loop system behavior, and the equilibrium of each state variable, respectively. According to these figures, it can be seen that if there is no control action, cell populations will not reach their state of equilibrium and the system is susceptible to parametric variation that could generate diabetes.

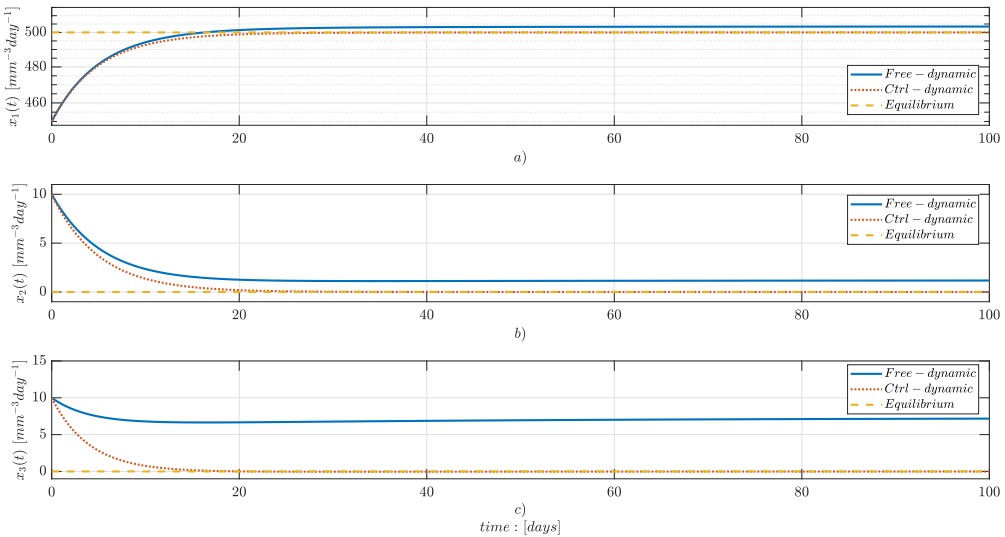

**Figure 3.** Convergence of the cell populations of resting macrophages, activated macrophages, and antigens to their equilibrium state. (**a**) $x_1$ to $x_1^*$. (**b**) $x_2$ to $x_2^*$. (**c**) $x_3$ to $x_3^*$.

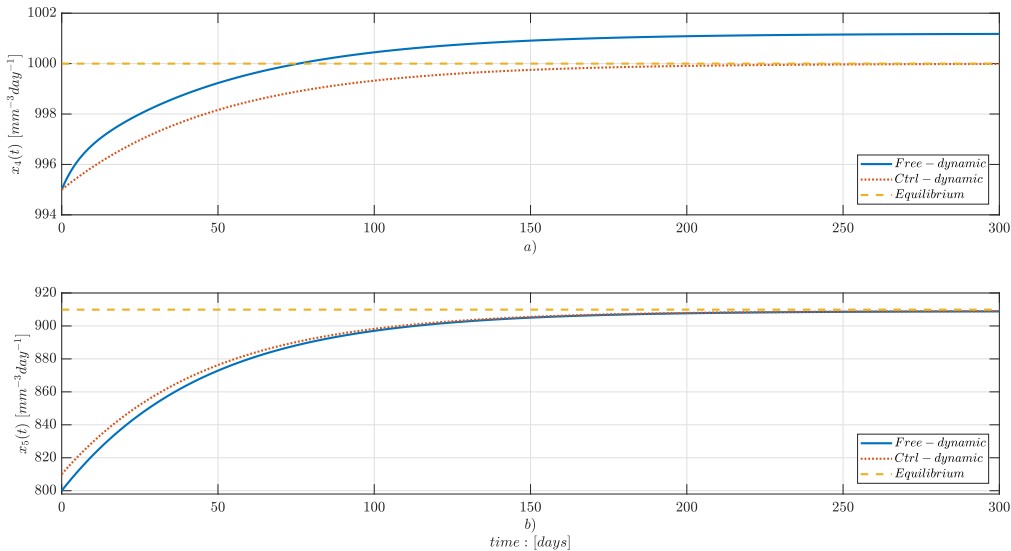

**Figure 4.** Convergence of the cell populations of T-cells and $\beta$-cells to their equilibrium state. (**a**) $x_4$ to $x_4^*$. (**b**) $x_5$ to $x_5^*$.

## 4. Discussion

A mathematical analysis of a nonlinear model aimed at understanding the complex relation between $\beta$-cells with the immunological response through autolytic T-cells and macrophages demands

a deeper understanding of biological concepts related to T1DM. The existence of a BPID, considering all upper bounds for the system state variables, gives conditions to establish a better understanding of the correlation between cells and the evolution of T1DM. According to the literature, in T1DM there is an inverse correlation between β-cells and autolytic T-cells, implying that when the autolytic T-cell population increases, the β-cell population decreases, activating a diabetic condition. Through the LCIS obtained in this study, cell populations can increase or vice versa. However, they stay limited within minimum and maximum population levels. At this point, we note that in a steady-state condition, cell populations never reach their state of equilibrium (see Figure 2), some of them even tend to their upper or lower bounds, such as the case of resting macrophages, activated macrophages, and autolytic T-cells, while an increase of β-cells is due to T-cell population decreases. Nevertheless, as can be seen in Figure 2c,d, the β-cells will not be greater in number than the autolytic T-cells because β-cells' upper bound becomes the autolytic T-cells' lower bound; at most, these populations would be equal. Therefore, it is vital to notice that autolytic T-cells cannot be zero in any case, making it so that a diabetic condition could appear at any time. The above emphasize the importance of a closed-loop control system able to compensate the proliferation of autolytic T-cells against the death rate of β-cells. Additionally, in this study we propose a closed-loop control system, and as can be seen in Figures 3 and 4, the proliferation of these cells is stabilized to their state of equilibrium, decreasing the rate in which macrophages become activated due to the interaction with antigenic proteins, diminishing the death rate of β-cells by the cell–cell interaction with autolytic T-cells as well as the proliferation of T-cells due to the profile of cytokines and chemokines induced by activated macrophages. We have proven that the control inputs, in accordance with the mathematical analysis, can keep cell populations at a stable level towards diabetes control in early diagnostics. However, nonlinear controller design for biological models entails more complex analysis compared with other nonlinear systems such as electric, mechanical, or chaotic. It is complex to analyze because the cell population sets cannot be directly controlled; this is only possible by manipulating some of their parameters. The feasibility of immunotherapy treatment requires a stable protocol based on clinical experimentation that supports mathematical hypotheses or suggestions, in which this research contributes to giving a basis for nonlinear control, with the corresponding biological implications, that may provide theoretical support for some clinical experiments related to this topic.

## 5. Conclusions

The assumption of considering autolytic T-cells as an invariant plane implies the existence of an input treatment that delays the proliferation of these cells due to activated macrophages, reducing the antigen population of β-cells; as a consequence, all dynamics can converge to the equilibrium point asymptotically.

The mathematical analysis suggests three control inputs that are directly related to the state variables: activated macrophages (24), antigens (25), and autolytic T-cells (26). In this case, control input (24) implies the existence of a counterpart that has a direct impact on avoiding any activation of macrophages by assuming that there is no antigen identified as a threat. The control input (25) aims at counter resting the antigen factor associated with β-cells based on the straight relation between the activated macrophages and the autolytic response. The last control input (26) is associated with holding the autolytic T-cell population that has a direct effect on reducing the β-cell population due to the influence of a higher cytotoxic cell of the immune response. Therefore, a mathematical analysis considering control-input based on a closed-loop provides a theoretical basis that leads to a more in-depth search aiming for an immunotherapy treatment that can be a reinforcement to actual procedures.

It is essential to acknowledge that the stimulation and propagation of diabetes is a combination of events, and there is no single event that is responsible for it. Simulations suggest that mathematical analysis on the search for upper bounds, as they are presented in Table 2, implies that a free parameter

such as (9) leads to the establishment of different assumptions in which upper and lower bounds for the autolytic T-cells are achieved (16).

In nonlinear controller design, the stability condition (30) represents a free parameter in which a high value of macrophage death rate and macrophage deactivation rate helps to maintain the stability of the cell populations. Hence, these parameters represent a suppression condition in which macrophages interact with the autolytic T-cells, giving a leading path on research in order to a deepen our understanding towards establishing an immunotherapy treatment. Therefore, the existence of a strategy that decreases the value of parameters $g$, $l$, $q$, and $s$, while increasing $c$ and $k$, could give positive results towards the control of diabetes in early diagnostics.

**Author Contributions:** Formal analysis, D.G.; Investigation, P.J.C.; Writing—review & editing, C.E.V. All authors have read and agreed to the published version of the manuscript.

**Funding:** This research received funding by the title project: Diseño de controladores y observadores no lineales en modelos relacionados a diabetes Mellitus Insulinodependiente, by Tecnológico Nacional de México/Instituto Tecnológico de Tijuana.

**Conflicts of Interest:** The authors declare no conflicts of interest.

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
