# Peer review of "Nonlinear Analysis for a Type-1 Diabetes Model with Focus on T-Cells and Pancreatic β-Cells Behavior"

_mca, doi:10.3390/mca25020023_

Round 1

Reviewer 1 Report

The paper states a hypothetical possibility to treat the diabetes type 1, preventing that the beta cells are destroyed by the immunological system. The authors analyze a nonlinear model relating such dynamics and found that if three controls are applied, the destruction of beta cells could be stopped. However, although the mathematical preliminary analysis is sound (except few details) is not deep, and the relationship with physiological aspects corresponding the possible treatment generated is absent. In this actual form, the paper cannot be accepted to be published, it needs further revisions.

  1. Please, re write the Introduction focusing on the open problems, and the contributions of the paper.
  2. In section 3, why is it proposed three controls?. One control is not enough?. In this sense, please compute the accessibility of the nonlinear model, and demonstrate that three inputs are required.
  3. The Direct Lyapunov method helps to determine the stability of a system f(x) around of p which fulfills that f(p)=0. The Lyapunov function then is constructed ensuring that V(p)=0, and V(x)>0. Please verify your Lyapunov function and the assumptions.
  4. Clarify the algebra to obtain the three controls in equations (23-25).
  5. The art-work of the paper is very bad. There is no units, and the simulations do not show much about the problem. Re think the figures.
  6. In the Discussion, give more physiological sense of the inputs, and discuss if such possible treatment could be realistic.

Author Response

LIST OF MAIN CHANGES ON THE MANUSCRIPT

First of all we want to thank you for your valuable feedback and accordingly with them, we did the next adjustments:

  1. Please, re write the Introduction focusing on the open problems, and the contributions of the paper.
  • Answer: Changes of the state of art were made. There were added more important references, they can be seen, in bold, throughout section 1: introduction.
  1. In section 3, why is it proposed three controls? One control is not enough? In this sense, please compute the accessibility of the nonlinear model, and demonstrate that three inputs are required.
  • Answer: These questions were responded in section 3. It is broadly discussed in section 3, in lines 266-275, also in lines 281-289 and 346-354. Here, it is described why there are proposed three control inputs according with biological implications. It is important to mention that the model under analysis have no control entries and we suggest these control inputs based on biological implications and their feasibility.
  1. The Direct Lyapunov method helps to determine the stability of a system f(x) around of p which fulfills that f(p)=0. The Lyapunov function then is constructed ensuring that V(p)=0, and V(x)>0. Please verify your Lyapunov function and the assumptions.
  • Answer: The results were verified accordingly with your comments. In section 2 were stablished stability conditions for the equilibrium point. Furthermore, this section was rewritten highlighting biological implications.
  1. Clarify the algebra to obtain the three controls in equations (23-25).
  • Answer: This suggestion was attended and more details about how these control inputs were proposed, see section 2 equations (23)-(31).
  1. The art-work of the paper is very bad. There are no units, and the simulations do not show much about the problem. Re think the figures.
  • Answer: We worked on each Figure such as units, labels, etc. in order to give a better understanding, see Figures 1 (page 10), 2 (page 11), 3 and 4 (page 12), section 3.
  1. In the Discussion, give more physiological sense of the inputs, and discuss if such possible treatment could be realistic.
  • Answer: As was explained in point 3, the entire work was focused on the biological sense of the proposal and it can be seen throughout these sections.

Best regards, thank you.

Reviewer 2 Report

Review of article entitled ‘Nonlinear analysis for a Type-1 diabetes model focus on T-cells and pancreatic β-cells behaviour’

The authors analyse a mathematical model in the form of ODEs to describe TIDM.

I recommend that the article is accepted for publication after incorporating these minor comments:

  1. Change the title to read ‘Nonlinear analysis for a Type-1 diabetes model with focus on T-cells and pancreatic β-cells behaviour’
  2. Re-write the sentence in the abstract on line 10 to 14: beginning ‘Therefore….’It is unclear. It did not make sense to me.
  3. On page 1, line 16, in the abstract. Change ‘point free’ to read ‘free point’
  4. In the Introduction on Page 1, Line 22 change ‘T-cells when’ to read ‘T-cells which’
  5. On Page 2, Line 35, change ‘can be find’ to read ‘can be found’
  6. On the same line 35, change ‘where is’ to read ‘where it is’
  7. On Page 2, Line 64, Change the sentence ‘The remainder of this work is organised with the following sections’ to read ‘The article is organised as follows’
  8. On Page 2, Line 74, change ‘modelled by a fifth order differential equations’ to ‘modelled by a system of ordinary differential equations’
  9. Perhaps explain why the model terms are all of exponential form. Do all the interactions happen for ever? Is there no saturation? Was it a simplifying assumption?
  10. Page 3, Line 97: Did you mean ‘Let us define’ instead of ‘Lest defined’
  11. Page 4, Line 111, change ‘lets defined’ to ‘define’
  12. Page 4, Line 120:change the statement ‘and the set of eigenvalues are located in the negative orthant’ to read ‘and the set of eigenvalues, for the given parameter values in Table 1, are located in the negative orthant’
  13. Page 10: Reduce the fonts on the vertical axes of Figures 2 and 3. That is the word ‘population’ should be reduced to not exceed the figure height.

Author Response

LIST OF MAIN CHANGES ON THE MANUSCRIPT

First of all we want to thank you for your valuable feedback and accordingly with them, we did the next adjustments:

  1. Change the title to read ‘Nonlinear analysis for a Type-1 diabetes model with focus on T-cells and pancreatic β-cells behavior’
  • Answer: This consideration is attended, and the title was rewrite as it is proposed, see title.
  1. Re-write the sentence in the abstract on line 10 to 14: beginning ‘Therefore….’It is unclear. It did not make sense to me.
  • Answer: The paragraph 10 to 14 was rewrite and changed, given the new paragraph giving in lines 8 to 19.
  1. On page 1, line 16, in the abstract. Change ‘point free’ to read ‘free point’
  • Answer: All the text between these lines was changed and this sentence results no necessary, therefore, it was eliminated.
  1. In the Introduction on Page 1, Line 22 change ‘T-cells when’ to read ‘T-cells which’
  • Answer: This comment was attended, and it can be seen in line 26.
  1. On Page 2, Line 35, change ‘can be find’ to read ‘can be found’
  • Answer: The sentence ‘’can be find’’ was eliminated and changed to ''there are many studies related to this topic'', see line 53.
  1. On the same line 35, change ‘where is’ to read ‘where it is’
  • Answer: Many of the text in this paragraph was changed and that word was removed, se lines 53-55.
  1. On Page 2, Line 64, Change the sentence ‘The remainder of this work is organized with the following sections’ to read ‘The article is organized as follows’
  • Answer: This suggestion was attended, literally, see line 100.
  1. On Page 2, Line 74, change ‘modelled by a fifth order differential equations’ to ‘modelled by a system of ordinary differential equations’
  • Answer: This suggestion was attended, literally, see lines 109 and 110.
  1. Perhaps explain why the model terms are all of exponential form. Do all the interactions happen forever? Is there no saturation? Was it a simplifying assumption?
  • Answer: In the paper is added three references labeled as [24]-[26] that review some related models to Diabetes Mellitus Type 1 based on ordinary differential equations; the analyzed model in this work was not defined or modeled-design by any of the authors. We took the reported model in literature with the respective parameter values [18], to implement the compact invariant domain method since all variables state of the model are located in the non-negative orthant due to biological implication that involves two complex central dynamics: beta-cells and the Leucocyte population. The model system under analysis begins with the premise of the misleading tag of the macrophages to pancreatic cells as an innate response. Consequently, the beta-cell population represents the prey as the prey-depredator scheme holds. Fifth-order ordinary differential equations describe the relation of the beta-cells (x5) in the presence of macrophages(x1), active macrophages(x2), and autolytic T-cells(x4), three main variables of the innate response of the immunological system; The discussed variable associated to the last variable: antigen(x3); is consequently a natural reaction of the prey-depredator system when beta-cells labeling as unwelcome populations. In our work, we are interested in analyzing the mathematical model when active macrophages and autolytic T-cells begin a communication chain that activates the Lymphocytes T-helper cells (T-CD4) to produced mass proliferation of the Lymphocytes T-cytotoxic cells (T-CD8). Therefore, the hypothesis held in this work is related to the idea that suppression or null activation by autolytic T-cells and active macrophages is a leading path to alternative immunotherapy, although at the moment is only presented the mathematical analysis with no clinical data, in-vitro study or in-silico trials based on experimental data. Correction in the document was done.
  1. Page 3, Line 97: Did you mean ‘Let us define’ instead of ‘Lest defined’
  • Answer: This sentence was changed to “Now, considered a nonlinear system represented as follows”, see lines 133 and 134.
  1. Page 4, Line 111, change ‘lets defined’ to ‘define’
  • Answer: This word was changed even the paragraph presents important changes, see line 146.
  1. Page 4, Line 120: change the statement ‘and the set of eigenvalues are located in the negative orthant’ to read ‘and the set of eigenvalues, for the given parameter values in Table 1, are located in the negative orthant’
  • Answer: These lines were removed, and it is cited the work where they are proposed.
  1. Page 10: Reduce the fonts on the vertical axes of Figures 2 and 3. That is the word ‘population’ should be reduced to not exceed the figure height.

• Answer: We worked on each Figure such as units, labels, etc. in order to give a better understanding, see Figures 1 (page 10), 2 (page 11), 3 and 4 (page 12), section 3.

Best regards, thank you.

Round 2

Reviewer 1 Report

The authors improve the paper and answer all the questions.

I have no more comments.

Author Response

Thanks for all your comments, we really appreciate them.

Reviewer 2 Report

All suggestions have been attended to. I recommned that the article is accepted for publication.

Author Response

(The authors gave the same response as above.)
